# Prognostic Immune Effector Signature in Adult Acute Lymphoblastic Leukemia Patients Is Dominated by γδ T Cells

**DOI:** 10.3390/cells12131693

**Published:** 2023-06-22

**Authors:** Anne-Charlotte Le Floch, Marie-Sarah Rouvière, Nassim Salem, Amira Ben Amara, Florence Orlanducci, Norbert Vey, Laurent Gorvel, Anne-Sophie Chretien, Daniel Olive

**Affiliations:** 1Equipe Immunité et Cancer, Centre de Recherche en Cancérologie de Marseille (CRCM), INSERM U1068, CNRS UMR7258, Institut Paoli-Calmettes, Aix-Marseille Université, UM 105, 13009 Marseille, France; leflocha@ipc.unicancer.fr (A.-C.L.F.);; 2Plateforme d’Immunomonitoring, Institut Paoli-Calmettes, 13009 Marseille, France; 3Département d’Hématologie, CRCM, INSERM U1068, CNRS UMR7258, Institut Paoli-Calmettes, Aix-Marseille Université, UM 105, 13009 Marseille, France

**Keywords:** acute lymphoblastic leukemia, γδ T cells, Vδ2 T cells, prognosis

## Abstract

The success of immunotherapy has highlighted the critical role of the immune microenvironment in acute lymphoblastic leukemia (ALL); however, the immune landscape in ALL remains incompletely understood and most studies have focused on conventional T cells or NK cells. This study investigated the prognostic impact of circulating γδ T-cell alterations using high-dimensional analysis in a cohort of newly diagnosed adult ALL patients (10 B-ALL; 9 Philadelphia^+^ ALL; 9 T-ALL). Our analysis revealed common alterations in CD8^+^ T cells and γδ T cells of relapsed patients, including accumulation of early stage differentiation and increased expression of BTLA and CD73. We demonstrated that the circulating γδ T-cell signature was the most discriminating between relapsed and disease-free groups. In addition, Vδ2 T-cell alterations strongly discriminated patients by relapse status. Taken together, these data highlight the role of ɣδ T cells in adult ALL patients, among whom Vδ2 T cells may be a pivotal contributor to T-cell immunity in ALL. Our findings provide a strong rationale for further monitoring and potentiating Vδ2 T cells in ALL, including in the autologous setting.

## 1. Introduction

The success of immunotherapy in acute lymphoblastic leukemia (ALL) has highlighted the critical role of the immune microenvironment in this disease [1,2]. Despite the advances in allogeneic hematopoietic stem cell transplantation (HSCT) and the use of immunotherapeutic approaches such as monoclonal antibodies and CAR T-cell therapy [3], most adult patients with ALL relapse [4,5]. Before the age of 55 years, overall survival (OS) rates range from 50 to 60%, while survival in older patients does not exceed 30% [6].

ALL are heterogeneous entities arising from B-cell (B-ALL) or T-cell (T-ALL) precursors. B-ALL accounts for approximately 75% of ALL cases. Of these, 25–30% harbor the BCR-ABL1 fusion protein, which is the driver mutation of Philadelphia chromosome-positive acute lymphoblastic leukemia (Ph^+^ -ALL). The prognosis of ALL is influenced by disease-related factors (central nervous system involvement, white blood cell count, cytogenetic and molecular subtype) and quality of response to treatment (MRD; Minimal Residual Disease), which are currently used for risk stratification and therapeutic decision making [3]. Although immunotherapeutic approaches predominate in ALL, immune changes remain poorly understood. Indeed, some studies have focused on cellular immune escape in ALL, demonstrating immunosuppressive functions of myeloid-derived infiltrating cells such as mesenchymal stromal cells (MSCs) [7], myeloid-derived suppressor cells (MDSCs) [8], and tumor-associated macrophages (TAMs) [9], or the antitumor role of non-classical monocytes [10]. Conversely, the contribution of effector immune cells has been less studied in ALL. Cytotoxic T lymphocyte (CTL) and natural killer (NK) cell infiltration show defective activation and function [11,12,13]. This poor immunogenic response has been proposed to be related to the low mutational burden in ALL [2], qualitative or quantitative defects in HLA expression [14], and an imbalance in the expression of co-stimulatory or inhibitory molecules (PD-1/PDL-1 pathway, CTLA-4, or TIM-3) [15,16,17]. Immune escape in ALL is also mediated by regulatory T cells (T-regs) [18] and their frequency has been correlated with response to immunotherapies [19,20].

Most immunomonitoring studies of immune effectors in ALL have so far focused on conventional T cells or NK cells but data on γδ T-cell populations are currently scarce. The crucial role of γδ T cells in antimicrobial and antitumoral immunity has recently been demonstrated [21]. In contrast to MHC-restricted αβ T cells [22], γδ T-cell activation is mediated by ligand recognition by both TCR and non-TCR receptors such as DNAM-1 [23] or NKG2D [24]. γδ T cells also represent a major discovery at the interface of innate and adaptive immunity [25,26]. In addition to their unique recognition mechanism, which remains partially identified, γδ T cells could mediate a rapid and potent cytotoxic response [27]. They can activate αβ T cells and invariant natural killer T cells and interact with NK cells, dendritic cells (DCs), macrophages, and neutrophils [28]. These properties make γδ T cells prime candidates for cancer immunotherapy [29,30]. Human γδ T cells represent 5–10% of circulating lymphocytes [31]. They can be classified into four groups based on the TCR δ-chain (Vδ1, Vδ2, Vδ3, and Vδ5) [32]. Human γδ T cells in tissues and peripheral blood are predominantly composed of Vδ1 and Vδ2 T cells. Both exhibit pleiotropic antitumoral responses but also have specific characteristics that depend on the cancer subtype [33,34]. Vδ2 T cells are the major blood γδ T-cell subtype and their activation is mediated by the recognition of non-peptidic phosphorylated metabolites that bind to butyrophilin 3A1 and 2A1 molecules [35,36,37,38].

In ALL, preliminary in vitro studies have shown that γδ T cells exert potent cytotoxic functions against primary ALL blasts [39,40,41,42]. In vivo experiments have confirmed the activity of γδ T cells in a xenogeneic CML model [43] and in a Ph^+^ ALL model [44]. γδ T-cell-based immunotherapy after HSCT has also shown promising results. Indeed, an increased frequency of ɣδ T cells in ALL patients undergoing αβ-depleted HSCT was associated with improved OS [45,46,47]. In addition, in vivo stimulation of Vδ2 T cells with zoledronate (ZOL) after haploidentical HSCT improved outcomes in pediatric ALL patients [48]. Several clinical trials are currently investigating γδ T-cell-based therapeutic approaches in various malignancies but few include ALL patients and only in the allogeneic setting [49]. Similarly, the prognostic role of γδ T cells in adult ALL has mainly been studied after HSCT, where a better reconstitution of γδ T cells was associated with a lower risk of relapse [45,46]. The role of γδ T cells at diagnosis of adult ALL is based only on bulk transcriptome inference, which showed that higher TCRVδ2^+^ γδ TILs (tumor-infiltrating lymphocytes) were associated with prolonged survival [50]. Qualitative changes in γδ T cells at diagnosis have only been investigated in small pediatric cohort studies of B-ALL, which showed that increased CTLA-4 and decreased CD8 expression on γδ T cells were associated with poor outcomes [51,52]. Importantly, most studies of γδ T cells in ALL have not distinguished between Vδ1 and Vδ2 subsets.

To date, no study has accurately described γδ T-cell subsets and their changes at diagnosis in adult ALL patients and analyzed their correlation with outcomes.

This study aims to investigate the prognostic impact of the γδ T-cell immunophenotype using high-dimensional analysis in a cohort of newly diagnosed treatment-naïve adult ALL patients. We mapped the peripheral gamma delta landscape by mass cytometry and determined the individual prognostic signature of Vδ1 and Vδ2 T cells.

## 2. Materials and Methods

### 2.1. Clinical Samples

Heparinized blood from 28 ALL patients were obtained before induction chemotherapy (10 B-ALL; 9 T-ALL; 9 Ph^+^ ALL), from the Department of Hematology of the Institute Paoli–Calmettes. Informed consent was obtained from all donors in accordance with the Declaration of Helsinki, and the study was approved by our Institutional Review Board (BTN-LAL-IPC2021-049—Immunomodulation dans les leucémies aigues—22 June 2021). Baseline characteristics of the ALL patients are shown in Appendix A. Patients were diagnosed between May 2005 and March 2019 and were aged from 18 to 80 years old. All patients were treated with conventional induction chemotherapy.

Samples with more than 70% of blasts were selected for the study.

### 2.2. Flow and Mass Cytometry

For flow cytometry, PBMCs were washed and incubated with human Fc Block (BD Biosciences, San Jose, CA, USA) for 10 min at 4 °C before immunostaining with a mix of extracellular antibodies. Acquisition was performed on a BD FACS CANTO II (BD Biosciences) and data were analyzed using DIVA 8.0.1. The mass cytometry experiments were performed as previously described [53]. Briefly, PBMCs were washed with RPMI 1640 medium supplemented with 10% fetal calf serum (FCS) and incubated for 1 h at 37 °C with 5% CO_2_ in RPMI 1640 with 2% FCS; Pierce Universal Nuclease 2.5 kU (Thermo Fisher Scientific, Waltham, MA, USA) was added in the last 30 min. The cells were centrifuged and incubated with cisplatin 5 µM to stain dead cells and then incubated with human Fc Block (BD Biosciences). A total of 1- to 2-million PBMCs were stained with the extracellular antibodies for 1 h at 4 °C (Appendix A, upper part). After centrifugation, the cells were washed and permeabilized with the Fixation/Transcription Factor Staining Buffer Set (eBioscience, San Diego, CA, USA) for 30 min at 4 °C. Cells were then preincubated with human Fc Block for 10 min at 4 °C before incubation with intracellular antibodies for 30 min at 4 °C (Appendix A, lower part). The cells were then washed and labeled overnight with 125 µM of DNA intercalator diluted in 2% PFA (Fluidigm-Standard Biotools, San Francisco, CA, USA). Finally, cells were diluted in EQ™ Four Element Calibration Beads (Fluidigm) and were acquired on a Helios mass cytometer (Fluidigm). After data acquisition, cells were further analyzed using FlowJo v10.6.2 (BD Biosciences) and OMIQ software from Dotmatics (accessed on 12 January 2023; www.omiq.ai, www.dotmatics.com).

### 2.3. Uniform Manifold Approximation and Projection Analysis

The uniform manifold approximation and projection (UMAP) dimensionality-reduction technique was used for immune subset identification and was performed using the OMIQ software from Dotmatics. Leukemic cells and immune subsets were identified according to the gating strategy shown in Appendix A. BGA and statistical analyses were performed when all immune cell subsets contained more than 30 cells. Hierarchical clustering (Euclidean distance) and heat map visualization were performed using Phantasus v1.19.3.

### 2.4. Generation of Anti-Human BTN3A mAb

To generate anti-BTN3A 20.1, BALB/c mice were immunized with a soluble BT3.1-Ig fusion protein, as previously described [54].

### 2.5. Expansion of Vδ2 T Cells

PBMCs from ALL patients and HV (healthy volunteers) were isolated by density gradient (Lymphoprep) and were then frozen until use. To expand autologous Vδ2 T cells, frozen PBMCs from ALL patients were stimulated with ZOL 1µM (Sigma-Aldrich, Saint Louis, MO) and rhIL-2 (Miltenyi Biotec, Bergisch Gladbach, Germany) at day 0. From day (d) 5, rhIL-2 was renewed every 2 days and the cells were maintained at 1.5 × 10^6^/mL until d14. rhIL-15 (10 ng/mL; Miltenyi Biotec) was also added starting at d2 and was renewed every 2 days, as IL-15 has been shown to enhance the proliferative capacity of Vδ2 T cells [55,56,57]. Fresh autologous expanded Vδ2 T cells were then used for functional assays, depending on the quantity of Vδ2 T cells. The fold increase in viable Vδ2 T cells was calculated using the formula: (d14 %Vδ2 × d14 total cell number/(d0 % Vδ2 × d0 total cell number).

### 2.6. Degranulation Assays

To analyze CD107 expression, fresh autologous Vδ2 T cells and primary ALL blasts were cocultured at an Effector:Target (E:T) ratio of 1:1 with anti-CD107a, anti-CD107b, Golgi stop, and anti-BTN3A 20.1 agonist mAb or isotype control (1 µg/mL). After 4 h, cells were collected and analyzed by flow cytometry. Cells were acquired on a BD FACS CANTO II and analyzed using DIVA 8.0.1.

### 2.7. Statistics

Data were analyzed using between-group analysis (BGA). Variables with a mean frequency expression of less than 5% were excluded to avoid any assessment of background noise. The BGA was performed using RStudio v2022.07.2 (made4 package). R scripts used in the BGA are provided in Appendix A.

Statistical analyses were performed using Graph Pad Prism 5.0 (Graph Pad Software, San Diego, CA, USA).

Normality of distributions was assessed using the D’Agostino–Pearson normality test.

Comparisons of continuous variables between two groups were performed using a two-tailed Mann–Whitney U test. The Kruskal-Wallis test followed by Dunn’s post hoc test was used for multiple comparisons of independent samples.

## 3. Results

### 3.1. γδ T-Cell Phenotypic Variables Are Significant Contributors to the Immune Effector Signature Associated with Relapse in ALL Patients

We first evaluated the global distribution of immune markers on circulating CD8^+^ T cells, NK cells, and γδ T cells, of adult ALL patients at diagnosis and compared their respective contribution according to clinical outcomes. We analyzed peripheral blood mononuclear cells (PBMCs) by mass cytometry from 25 analyzable patients (9 B-ALL, 9 B-ALL Ph^+^, and 7 T-ALL) and quantified immune markers by manual gating (Appendix A). This resulted in 89 variables that were analyzed by a between-group analysis (BGA) to generate a composite immune signature discriminating relapsed (REL) from disease-free (DF) ALL patients (n = 15 and n = 10, respectively) (Appendix A). The BGA analysis provides the contribution of each immune variable and the relative contribution of each immune cell type to the discrimination of the groups compared. Of these 89 variables, 32, 28, and 29 were related to γδ T cells, CD8^+^ T cells, and NK cells, respectively; therefore, the random theoretical relative contribution of each immune cell subset to group discrimination was 35.9%, 31.5%, and 32.6%, respectively. Figure 1A shows the result projection of all samples from the BGA analysis. The relative contribution of each variable according to the immune cell subtype is shown in Figure 1B. Variables associated with γδ T cells contributed the most to the global discrimination (43.3%), followed by CD8^+^ T cells (33.2%), and NK cells (23.5%). By subtracting the expected contribution from the total contribution, we found that γδ T-cell variables contributed +7.4% more than expected to the discrimination of groups, versus +1.7% for CD8^+^ T cells, and −9.1% for NK cell variables (Figure 1C).

Consistently, analysis of the 20 most discriminating variables revealed 11, 6, and 3 immune variables associated with ɣδ T cells, CD8^+^ T cells, and NK cells, respectively (Figure 1D). Variables associated with the differentiation and polarization states of both CD8^+^ T cells and ɣδ T cells were the most differentially enriched; thus, the frequencies of effector memory (EM) CD8^+^ T cells and EM or EMRA (effector memory T cells re-expressing CD45RA) ɣδ T cells were enriched in the DF group, whereas naive and CM ɣδ T cells were enriched in the REL group. Consistently, DF patients had increased cytotoxic and priming markers on CD8^+^ T cells (CD16^+^, CD56^+^) and ɣδ T cells (CD16^+^, CD57^+^, NKG2A^+^). In addition, some inhibitory or regulatory molecules were upregulated in the REL group: BTLA and CD73 on both CD8^+^ T cells and ɣδ T cells, and TIM-3 on ɣδ T cells. In contrast, the DF group was enriched for less mature NK cell subsets (CD56^dim^ CD16^−^ and CD56^dim^ CD16^+^CD57^−^), whereas highly differentiated NK cells (CD57^+^) were found in the REL group. Comparison of the frequencies of the 20 most discriminating variables revealed a significant difference between DF and REL patients only for the ɣδ T-cell variables; REL patients had increased naive ɣδ T cells, whereas DF patients had increased frequencies of EM ɣδ T cells and NKG2A^+^ ɣδ T cells (Figure 1E).

We also examined the differences in immune subsets according to ALL subtype; compared to HV, B-ALL patients had a lower frequency of monocytes. Conversely, T-ALL patients had a higher frequency of B-cells (Appendix A). Regarding ICI expression, ALL patients showed high heterogeneity with a global trend towards higher ICI levels in REL patients (Appendix A). TIM-3 was significantly upregulated on ɣδ T cells from REL patients and tended to be more expressed on CD8^+^ T cells and NK cells. A trend for other ICI upregulation was observed in REL patients, such as TIGIT on NK cells and BTLA and PD-1^+^ on ɣδ T cells and CD8^+^ T cells.

This BGA of circulating immune effector markers in 25 ALL patients revealed common immune alterations in both CD8^+^ T cells and γδ T cells and demonstrated that the circulating γδ T-cell signature was the most discriminant between relapsed and disease-free groups. These results raise the question of the respective contribution of the two most abundant circulating ɣδ T-cell populations: the Vδ1 and the Vδ2 subsets.

### 3.2. Prognostic Impact of ɣδ T-Cell Alterations in ALL Mainly Depends on Vδ2 T Cells

Next, we compared the different ɣδ T-cell subsets according to the occurrence of relapse. Figure 2 shows the results of UMAP analysis on γδ T cells from all ALL patients. Analysis of the density of the γδ T-cell subpopulations in each group (Figure 2A, left panel) showed marked differences in both Vδ1 and Vδ2 T-cell subsets but the frequency of γδ T cells among lymphocytes was not different between DF and REL patients. The frequencies of Vδ1 T cells, Vδ2 T cells, and of ɣδ^+^ Vδ2^−^ Vδ1^−^ T cells among γδ T cells were also similar (Figure 2B). We further investigated the expression of the most discriminating variables associated with ɣδ T cells according to our previous BGA analysis (Figure 2A, right panel).

Some UMAP changes were observed on Vδ1 T cells and Vδ2 T cells, such as a globally increased expression of CD27^+^ subsets in REL patients. In contrast, CD73 was expressed on Vδ1 T cells from REL patients and BTLA was predominantly expressed on Vδ2 T cells from REL patients (Appendix A). PhenoGraph clustering allowed the identification of 22 γδ T-cell subpopulations, according to the co-expression of 32 γδ T-cell variables (Appendix A). Consistent with previous results, a pronounced heterogeneity of cluster repartition was observed; however, some clusters displayed a similar pattern regardless of the ALL subtype. Thus, an increased proportion of NKG2A+ Vδ1 T cells (cluster 9) and a decreased frequency of cluster 3 corresponding to CM Vδ2 T cells (CD27+, CD28+, CD45RA−) expressing low levels of cytotoxic markers and high levels of PD-1 and BTLA, were observed in the DF group. Both Vδ1 T cells and Vδ2 T cells showed phenotypic changes associated with relapse. We therefore further investigated their own contribution by performing a BGA (Appendix A). The BGA of Vδ1 T-cell variables showed a clear sample dispersion (Figure 2C), which was more pronounced in the DF group. In contrast, most DF samples co-segregated in the BGA of Vδ2 T-cell variables (Figure 2F). Furthermore, non-hierarchical clustering (NHC) revealed that Vδ1 and Vδ2 T cells were segregated according to their differentiation state (Figure 2D,G). The naive population was associated with 4-1BB, OX-40, and CTLA-4 on both Vδ1 and Vδ2 T cells. In Vδ1 T cells, the EM population co-clustered with CD16 and NKG2A, and the EMRA population with GZMB and CD57, while frequencies of EM and EMRA Vδ2 T cells were associated with CD57 and CD8. In addition, the CM subset was associated with CD73 on Vδ1 T cells and with BTLA on Vδ2 T cells. Overall, the NHC of Vδ1 T-cell variables poorly discriminated patients by relapse status, whereas the NHC of Vδ2 T-cell variables identified distinct clusters of DF and REL patients.

Further analysis of the most discriminating Vδ1 and Vδ2 T-cell variables found in the BGA revealed a significantly increased frequency of EM subsets in both Vδ1 and Vδ2 T cells in DF patients (Figure 2E,H). In addition, Vδ1 T cells from the REL group tended to have decreased levels of NKp30 and increased levels of CD73, TIM-3, and CTLA-4. In contrast, Vδ2 T cells from REL patients had higher levels of Eomes and tended to have increased BTLA expression, fewer cytotoxic markers (CD8, CD16), and a lower expression of CD57 and NKG2A.

This comparative high-dimensional analysis of Vδ1 and Vδ2 T cells revealed similar characteristics of both subsets in REL patients, with a decreased frequency of the most differentiated subsets; however, Vδ1 and Vδ2 T cells also exhibited a distinct inhibitory and regulatory profile, including an increased frequency of CD73^+^ Vδ1 T cells and of BTLA^+^ Vδ2 T cells. Importantly, Vδ2 T-cell variables made an important contribution to the BGA analysis and a non-hierarchical method confirmed their association with prognosis. These findings suggest that within the ɣδ T-cell population, phenotypic alterations in Vδ2 T cells help to discriminate patients who will relapse, and also suggest that these Vδ2 T cells may have reduced effector capacities. To this end, we next investigated the extent to which these phenotypic changes were associated with functional alterations.

### 3.3. Vδ2 T Cells from Relapsed Patients Expand and Are Able to Degranulate and to Produce Th1 Cytokines

The expansion capacities of Vδ2 T cells were then explored with respect to the relapse status. Using ZOL stimulation combined with IL-2 plus IL-15 for 14 days, Vδ2 T cells from PBMCs of 18 ALL patients at diagnosis were expanded. REL patients had similar expansion capacities compared to DF patients (Figure 3A). Some of the samples were functionally tested by measuring degranulation and Th1 cytokine production of Vδ2 T cells, which did not appear to be altered in samples from REL patients, either spontaneously (Figure 3B) or by targeting BTN3A with the 20.1 agonist monoclonal antibody (Figure 3C).

This demonstrated, for the first time to our knowledge, that Vδ2 T cells from adult ALL patients were both able to expand in vitro after ZOL stimulation and to mediate cytotoxic activity against autologous blasts. It was also observed that the use of an anti-BTN3A agonist mAb enhanced the degranulation capacities of autologous Vδ2 T cells against primary ALL blasts, confirming results previously obtained only in primary AML samples [58].

Collectively, these results demonstrate that relapse-associated alterations in Vδ2 T cells do not affect either their proliferative or cytotoxic functions.

## 4. Discussion

While the prognostic value of ɣδ T cells has been studied in adult ALL patients treated with HSCT, nothing was known about their potential impact at diagnosis. Here, we provide the first high-dimensional profiling of blood γδ T cells in newly diagnosed adult ALL patients and show that their phenotypic changes discriminate patients according to clinical outcome. ɣδ T cells are the most discriminant cytotoxic cell subtype and show common alterations regardless of ALL lineage; therefore, our results confirm previous studies on the prognostic impact of ɣδ T cells in pediatric ALL patients [51,52], and provide a comprehensive analysis of Vδ1 and Vδ2 T-cell alterations in adult ALL patients.

The global immune effector signature between DF and REL patients identified major changes in conventional CD8^+^ T cells and γδ T cells. Some were shared by both immune cells, including in DF patients, the expansion of late-stage CD8^+^ T cells and γδ T cells, which was also associated with an increased expression of cytotoxic molecules, including CD16 on CD8^+^ T cells and γδ T cells, and CD56 on CD8^+^ T cells. T-cell maturation stage has not yet been associated with prognosis in ALL patients but an increased infiltration of late-stage effector T cells was found in the BM of 100 B-ALL patients compared to HV [17]. In addition, CD8^+^ T cells and γδ T cells from REL patients exhibited a globally immature and poorly cytotoxic profile, with increased levels of CD73 and BTLA. In contrast to the known immunosuppressive effect of CD73-adenosine signaling in solid tumors [59], there are no data on the CD73 expression on T cells in ALL; however, CD34^+^CD73^+^ blasts overexpress multidrug resistance markers such as BCL2, PGP, and MRP1 [60]. BTLA expression on TILs has been frequently associated with impaired anti-tumor T-cell responses in several cancer subtypes [61]. In large-cell lymphoma, BTLA^+^ T cells are less differentiated and have impaired killing capacity [62]. Regarding the impact of BTLA on ɣδ T cells, BTLA limits the proliferation and the cytokine secretion of mature lymph node γδ T cells [63], and our team has shown that BTLA decreases the proliferation of Vδ2 T cells and may also limit their differentiation [64]. In both adult AML and ALL, BTLA expression on primary blasts correlates with poor outcomes [65,66]. In addition, genetic deletion of BTLA in a mouse model of ALL is associated with impaired blastic cell proliferation and colony formation [66]. Taken together, BTLA and CD73 expression on both blastic and immune cells may have a negative impact on ALL. Accordingly, it can be hypothesized that the ALL microenvironment may affect both leukemic and non-leukemic cells in the same way as has been demonstrated in T-ALL [67]. Indeed, our findings reinforce the need to better understand the regulation of these molecules during leukemogenesis and their role in immune evasion mechanisms. Regarding other ICI, TIM-3 tended to be overexpressed on all immune effectors studied. These results are consistent with the emerging role of this novel immunosenescence marker in ALL. TIM-3 inhibits CD8^+^ T-cell responses in early T-ALL [67], and higher numbers of TIM-3^+^/PD-1^+^ CD4^+^ T cells or TIM-3^+^ CD4^+^ T cells predict poor survival in adult B-ALL [15] and pediatric B-ALL [17]. In addition, TIM-3^+^ ɣδ T cells have reduced killing capacity against colorectal cancer cells [68]. Furthermore, highly heterogeneous ICI expression was observed in our samples with a trend toward global upregulation. This is in line with previous studies showing a significant heterogeneity of PD-1, CTLA-4, or TIM-3 expression on CD8^+^ T cells in ALL patients compared to HV [15,17]. However, in contrast to BTLA and TIM-3, the prognostic contribution of other ICIs did not appear to be critical in our analysis. It can also be hypothesized that this heterogeneity reflects the diversity of our cohort; however, such heterogeneous signatures have been reported in both B-ALL [69] and T-ALL [70]. Indeed, analysis of ɣδ T-cell clusters had shown that some cluster changes were restricted to one ALL lineage, these data were consistent with the discrepancies in the immune landscape observed in T-ALL and B-ALL [71] and suggest that this may also be true for ɣδ T cells.

Our results showed that among circulating ɣδ T cells, the Vδ2 T-cell signature associated with relapse was more discriminant. Vδ2 T cells from REL patients had a poorly differentiated profile with low cytotoxic markers and increased BTLA and Eomes levels. In CD8^+^ T cells, Eomes promotes the acquisition of cytotoxic potential [72] but is also involved in the control of T-cell exhaustion [73]. Although Vδ2 T cells are absent in rodents, it has been shown that γδ T cells from mice expressing high levels of Eomes exhibit an exhausted phenotype and a reduced IFN-γ production [74]. Despite their marked alterations, Vδ2 T cells from REL patients were “activable”, as they were able to expand after ZOL treatment—similar to the DF ALL group—and mediate effector functions against autologous blasts. To date, the only γδ T cells described to expand in vitro in ALL are Vδ1 T cells [42]. Previously, expansion of Vδ2 T cells from PBMCs of ITK-treated CML patients with BrHpp or ZOL has been reported, and these cells efficiently kill ZOL-sensitized autologous or allogeneic CML cells [43]. As in other malignancies with bone involvement, ZOL represents an emerging treatment in ALL [75] and its pleiotropic effects may also be based on ZOL-induced activation of Vδ2 T cells. Indeed, our results support the successful in vivo expansion of Vδ2 T cells from 46 pediatric ALL patients following ZOL administration after TcRαβ/CD19-depleted haploidentical HSCT [48]. ZOL treatment resulted in the induction of both Vδ2 and Vδ1 T-cell subsets, with increased cytotoxicity of Vδ2 T cells against primary leukemia blasts [76].

Our study may have been limited by a small number of patients, and the effect of ICI expression on blood samples may not reflect the degree of T-cell exhaustion in the microenvironment. Tumor-infiltrating T cells are known to have a more exhausted phenotype than their circulating counterparts in both solid tumors [77,78] and in B-ALL after HSCT [14]; however, conflicting data have been reported for ɣδ T cells in AML and in MM where no differences in checkpoint expression on Vδ1 and Vδ2 subsets have been observed [79]. Further studies are warranted to confirm our findings in a prospective manner and to compare the value of immune markers in the BM and peripheral compartments in ALL and their respective characteristics according to the ALL subtype. Furthermore, γδ T cells are a major contributor to the efficacy of immune checkpoint blockade (ICB) in many cancers [80,81], reinforcing the importance of further investigating the impact of ICI on these immune effectors in ALL patients. Finally, ALL has been described to have the lowest cytolytic value compared to other hematologic malignancies [71]; thus, the high killing potential of ɣδ T cells combined with the observation of enriched cytotoxic subsets of ɣδ T cells in ALL patients with a good prognosis highlights the key role that ɣδ T cells may play.

## 5. Conclusions

Taken together, this high-dimensional analysis of phenotypic changes places ɣδ T cells at the center of ALL effector immunity. Our focus on ɣδ T cells has revealed that ALL patients with relapse have major ɣδ T-cell alterations at diagnosis and has confirmed that Vδ2 T cells may be a key contributor to the prognosis of ALL. Our findings provide a strong rationale for further monitoring and potentiating Vδ2 T cells in ALL, including in the autologous setting.

## Figures and Tables

**Figure 1 cells-12-01693-f001:**
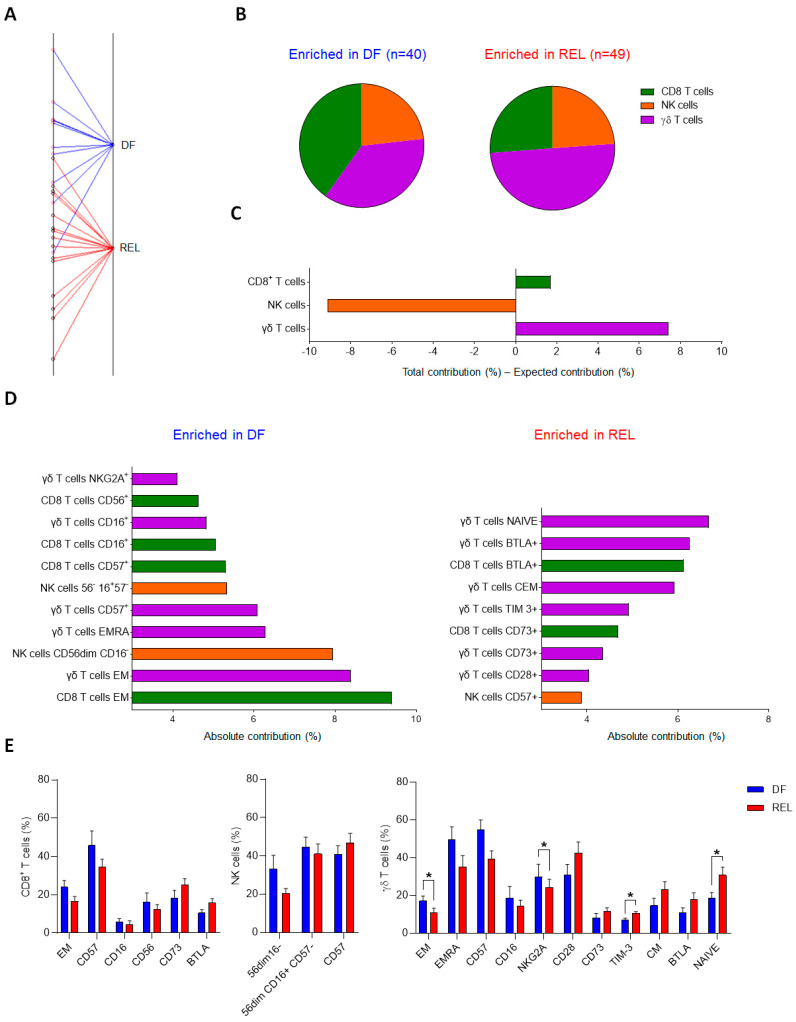
γδ T-cell phenotypic variables are significant contributors to the immune effector signature associated with relapse in ALL patients. Results of between-group analysis (BGA) using 89 variables associated with CD8+ T cells (n = 28), NK cells (n = 29), or γδ T cells (n = 32) from PBMCs of 9 B-ALL, 9 B-ALL Ph^+^, and 7 T-ALL patients are shown. (**A**) The axis shows the projection of all samples (n = 25). Sample origins from disease-free (DF) ALL patients (n = 10) are annotated in red; sample origins from relapsed (REL) ALL patients (n = 15) are annotated in black. Each sample origin is linked to its own group (blue trait for DF group; red trait for REL group). The degree of discrimination between groups and samples is given by the distances between group origins and the distances between sample origins, respectively. (**B**) The pie chart displays the respective contribution of the variables related to CD8+ T cells, NK cells, and γδ T cells, to the discrimination of the DF group (blue) from the REL group (red). (**C**) The bar graph, whose *Y*-axis shows the difference between the total contribution and the expected contribution of each population to the discrimination of the DF group from the REL group. (**D**) The contribution of the top 20 discriminating immune variables from the BGA. The variables enriched in DF patients are shown in the left panel and the variables enriched in REL patients are shown in the right panel. (**E**) Frequency comparison of the top 20 circulating immune variables discriminating REL vs. DF groups. Comparison of immune cell variables of CD8+ T cells and NK cells from 10 B-ALL, 9 B-ALL Ph^+^, and 9 T-ALL patients, between DF group and REL group (DF, n = 11; REL n = 17). Comparison of immune cell variables of γδ T cells from PBMCs of 9 B-ALL, 9 B-ALL Ph^+^, and 7 T-ALL patients, between DF group and REL group (DF, n = 10; REL, n = 15). Data are expressed as mean ± standard error of the mean (SEM) (**E**). The statistical significance was established using a Mann–Whitney test. * *p* < 0.05 (**E**).

**Figure 2 cells-12-01693-f002:**
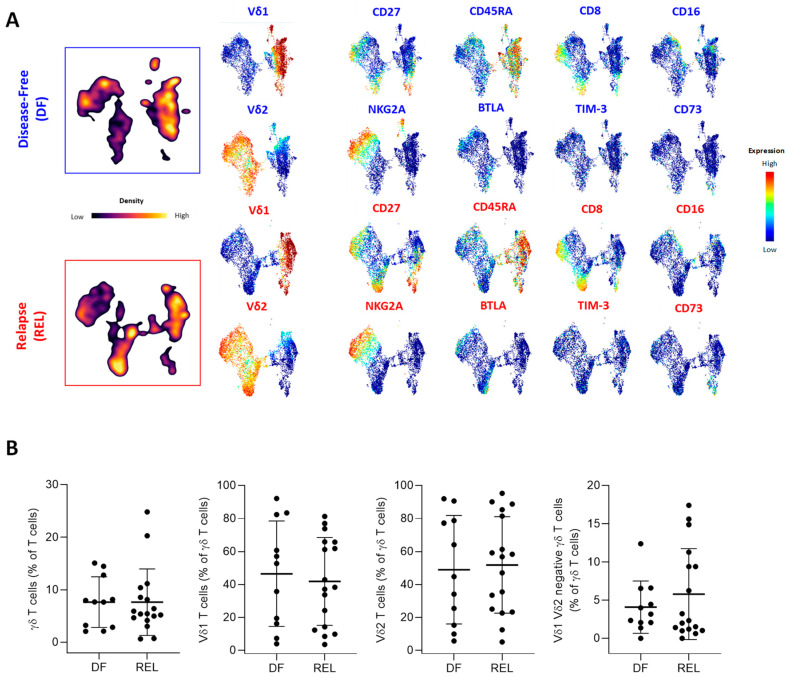
Prognostic impact of ɣδ T-cell alterations in ALL mainly depends on Vδ2 T cells. (**A**) Peripheral blood γδ T cells from ALL patients were manually gated, and consensus files were generated using the uniform manifold approximation and projection (UMAP) technique, with 5500 γδ T cells for each group (DF, REL). In the left panel, the density of γδ T-cell subpopulations in each patient group is projected (purple, low cell density; yellow, high cell density). Expression of markers of γδ T-cell maturation, cytotoxicity, costimulation, and coinhibition are projected on UMAP maps in the right panel (blue, low expression; red, high expression). (**B**) The proportion of ɣδ T cells among total T cells and the proportion of Vδ1^+^, Vδ2^+^, and Vδ1^−^/Vδ2^−^ T cells among total ɣδ T cells (%) according to relapse status. (**C**) The results of a BGA using 33 variables associated with Vδ1 T cells are shown. The axis displays the projection of all samples (n = 23). Sample origins from DF patients (n = 9) are annotated in red; sample origins from REL patients (n = 14) are annotated in black. Each sample origin is linked to its own group (blue trait for DF group; red trait for REL group). (**D**) Unsupervised hierarchical clustering of 33 Vδ1 T-cell variables used in BGA. (**E**) Top 10 discriminating Vδ1 T-cell variables from BGA analysis. (**F**) The results of a BGA using 33 variables associated with Vδ2 T cells are shown. The axis displays the projection of all samples (n = 23). Sample origins from DF patients (n = 9) are annotated in red; sample origins from REL patients (n = 14) are annotated in black. Each sample origin is linked to its own group (blue trait for DF group; red trait for REL group). (**G**) Unsupervised hierarchical clustering of 33 Vδ2 T-cell variables used in BGA. (**H**) Top 10 discriminating Vδ2 T-cell variables from BGA analysis. Data are expressed as mean ± SEM. The statistical significance was established using a Mann–Whitney test (**B**,**E**,**H**). * *p* < 0.05.

**Figure 3 cells-12-01693-f003:**
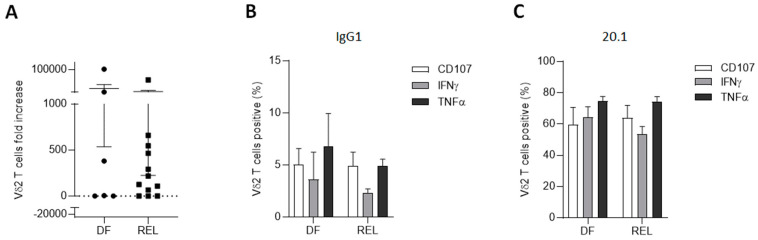
Vδ2 T cells from relapsed patients expand and are able to degranulate and produce Th1 cytokines. (**A**) PBMCs from ALL patients at diagnosis (n = 18; 6 DF and 12 REL patients) were cultured with ZOL for 14 days. (**B**,**C**) Effector functions of autologous expanded Vδ2 T cells (n = 4; 2 DF and 2 REL patients) were assessed after 4 h of co-culture with autologous primary ALL blasts (E:T ratio 1:1) in the presence of anti-BTN3A 20.1 (**C**) or its isotype control (**B**). Data are expressed as mean ± SEM. Statistical significance was determined by Mann–Whitney test (**A**).

## Data Availability

The datasets used and/or analyzed during the current study are available from the corresponding authors upon reasonable request (leflocha@ipc.unicancer.fr).

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
