# Peer review of "Prognostic Immune Effector Signature in Adult Acute Lymphoblastic Leukemia Patients Is Dominated by γδ T Cells"

_cells, 2023, doi:10.3390/cells12131693_

Round 1

Reviewer 1 Report

In this manuscript, authors have investigated the prognostic impact of Ƴδ T-cells phenotype in newly diagnosed adult ALL patients. The authors have designed the experiments scientifically and explained the results understandably. But the quality of the figures must be improved. For example, figure 1 is so poor that it is impossible to correlate it with text even after magnifying Figure 1. After the enhanced quality of Figure 1, only it can be reviewed. Rest all other sections are adequately explained, but the quality of figures should be improved in them.

In this manuscript, authors have profiled and analyzed the blood T cells (specifically detailed profiling of Ƴδ T-cells) in newly diagnosed adult ALL patients. This manuscript authors have observed the following significant findings which can add to the prognostic value in the case of adult ALL patients:

1. Phenotypic changes in Ƴδ T-cells profile discriminate ALL patients according to the clinical outcomes.

2. They have found that the Ƴδ T-cells are the most discriminating cell type and show common alterations regardless of the different ALL lineages.

3. They have shown that the relapsed adult ALL patients have a globally immature and poorly cytotoxic profile (like increased BTLA and CD73 levels) compared to DF patients.

4. Authors have shown that among the Ƴδ T-cells, Vδ2 T-cells in the relapsed cohort had poorly differentiated profiles with low cytotoxic markers. This study strengthens the need to understand better ICI molecules' regulation and their role in the immune evasion mechanism.

Even though the findings are of great importance, the following issues can be addressed:

1. Figure 1 must be improved significantly, as it's impossible to go through the details of molecules and data in Figure 1. Therefore, figure 1 must be replaced by a better-quality of the figure.

2. All the findings of this manuscript are based on the data of 25 patients (9 B-ALL, 9 B-ALL Ph+, and 7 T-ALL patients). If it is possible to extend this study to a larger cohort of patients, it will significantly increase the impact of this finding.

Author Response

Thank you very much for reviewing our manuscript and suggesting how to improve it, which we've tried to incorporate into this new version.

Regarding the general comments, some English language and style changes have been corrected, and the methods section has been implemented regarding patient selection criteria.

Regarding the quality of the figures, we have improved Figures 1 and 2 by significantly unifying and enlarging figures. It should be noted that there is an error in the number of patients in Figure 2 (23 and not 22 ALL samples were analyzed for Vδ1 and Vδ2 T cell immunophenotype).

For Figure 1, Panel A (strategy for identification of immune cell subsets and T cell subsets) has been moved to Supplementary figure1, and the waterfall plot has been changed into a pie chart to simplify the message.

In addition, as suggested by the other reviewer, we have removed the supplementary figures related to the statistical comparison between ALL subtypes, due to low power in the comparison groups.

Our study focused on rare cell subsets associated with a rare malignant hemopathy. These rare subsets are even rarer in blastic samples at diagnosis.

- The incidence of ALL in the United States is 1.8 per 100,000 persons per year (SEER), with adult ALL accounting for 20% of all ALL.

- We believe that our assay is optimized regarding sample availability and is consistent with previously published immunophenotypic data available in ALL, with similar effectiveness to our study (example: NK cells; Duault. Blood. 2021, n=20). In addition, it must be considered that the available data on γδ T cells in ALL do not exceed 19 pediatric B-ALL patients (Pawlik-Gwozdecka et al., Arch Med Sci 2021).

Mass cytometry provides an analysis at the single cell level, allowing interpretation of small sample sizes and study of rare cell subsets (Yao et al., J. Immunol. Methods 2014; Spitzer et Nolan, Cell 2016). For this reason, we have kept the part related to UMAP maps of Vδ1 and Vδ2 T cells by ALL lineage.

However, our results should be validated in a prospective cohort, as added in the Discussion.

Reviewer 2 Report

This paper investigated the prognostic impact of circulating γδ T cell alterations using high-dimensional 18 analysis in a cohort of newly-diagnosed adult ALL patients, including 10 B-ALL, 9 Philadelphia+ ALL, and 9 T-ALL. I have the following comments:

1. Although the paper conducted a comparison of profile of circulating cells according to ALL lineage, the sample size in each lineage group is very small, and this comparison would lack statistical power. As a result, not being able to detect a difference between lineages with this small sample size would not mean that there are truly no differences. With this small sample size, it can be difficult to assess to what extent the findings between relapse vs. non-relapsed specimens were due to heterogeneity of ALL subtypes.  

2. The analyses in the paper should adjust for ALL subtypes. 

3. How many relapses and non-relapses are there in each ALL subtypes? This information should be added to the paper. 

4. The sample size in this paper is overall small. Validation of the findings will be needed.

5. How were the 28 ALL patients selected for this study? It will be helpful to describe the selection criteria and procedure. Similarly, how were the healthy volunteers selected?

6. Resolution of some figures (e.g., Figure 1) is poor and can be improved.

Author Response

Thank you very much for the quality of your review and for your suggestions for improving our manuscript, which we've tried to incorporate in this new version.

Regarding the general comments, some English language and style changes have been corrected, and the methods section regarding patient selection criteria has been implemented.

Regarding the different points you raised:

1 and 2: The statistical power might indeed be questionable to compare such a small sample size from a statistical point of view. Therefore, we have removed the supplementary figures and the discussion related to the statistical comparison between ALL subtypes.

3: The respective numbers of REL and DF patients are indicated in the legend of  Supplementary Fig.4. Since the part concerning the comparison between ALL subtypes has been significantly reduced in this revised version, this information has not been added in the main text of this revised version.

4: Our study focused on rare cell subsets in the context of a rare malignant hemopathy. These rare subsets are even rarer in blastic samples at diagnosis.

- ALL affects 1.8 per 100,000 individuals per year in the United States (SEER), considering that adult ALL represent 20% of the total ALL.

- We believe that our effectiveness is optimized regarding sample availability and is in line with previously published immunophenotypic data available in ALL, with a similar number of patients. (Example: NK cells; Duault. Blood. 2021, n=20).  In addition, it must be considered that the available data on γδ T cells in ALL has not yet exceeded 19 pediatric B-ALL patients (Pawlik-Gwozdecka et al., Arch Med Sci 2021).

Mass cytometry provides an analysis at the single cell level and allows interpretation of small sample sizes and study of rare cell subsets (Yao et al., J. Immunol. Methods 2014; Spitzer et Nolan, Cell 2016). For this reason, we have kept the part related to UMAP maps of Vδ1 and Vδ2 T cells by ALL lineage.

However, our results should be validated in a prospective cohort, as added in the Discussion.

5: The selection criteria for ALL patients were added in the Methods section.

- Patient criteria:

* Newly diagnosed ALL

> 18 years of age

* Treated with induction therapy

- Sample criteria:

* 2 samples for each patient with > 20M thawed cells/vial (1st sample was used for phenotypic analysis +/- expansion and 2nd vial for possible functional analysis)

> 70% PB blasts (for functional assay)

Flowchart samples:

For HV, we selected samples from the EFS, which accepts only blood from HV under the age of 65. The median age of HV used in this study is 51 [19-64], which is quite similar to the cohort of ALL patients (48 [19-81]).

We have added more detailed information on the selection criteria of patients in the Methods section.

6: Regarding the quality of the figures, we have improved Figures 1 and 2 by significantly unifying and enlarging figures. It should be noted that there is an error in the number of patients in Figure 2 (23 and not 22 ALL samples were analyzed for Vδ1 and Vδ2 T cell immunophenotype).

For Figure 1, Panel A (strategy for identification of immune cell subsets and T cell subsets) has been moved to Supplementary Figure 1, and the waterfall plot has been changed into a pie chart to simplify the message.